# A qualitative study of knowledge, attitudes and perceptions towards malaria prevention among people living in rural upper river valleys of Nepal

Kiran Raj Awasthi⬚*, Jonine Jancey, Archie C. A. Clements, Justine E. Leavy

Curtin School of Population Health, Curtin University, Perth, WA, Australia

* kiran.awasthi@curtin.edu.au

**Data Availability Statement:** The data has been deposited in the qualitative data repository with the link below https://doi.org/10.5064/F6WMOBYB.

## Abstract

### Background

Nepal has made significant progress in decreasing the number of malaria cases over the last two decades. Prevention and timely management of malaria are critical for the National Malaria Program in its quest for elimination. The study aimed to explore the knowledge, attitudes and behaviour towards malaria prevention and treatment among people living in rural villages of Khatyad Rural Municipality in Nepal.

### Methods

This qualitative study collected information through virtual in-depth interviews (N = 25) with female and male participants aged between 15 and 72 years.

### Results

More than half of the participants knew about the causes of malaria, were aware of the complications of untreated malaria and knew that anti-malarial medicines were provided for free at the public health facilities. Participants indicated that their first choice of health care were public health facilities, however limited supply of medications and diagnostics deviated patients to the private sector. While tertiary care costs were not financially viable, participants opted against traditional care for malaria. Factors such as cost of treatment, distance to the health facility and the decision making authority in households influenced health related decisions in the family. Although long-lasting insecticidal nets were distributed and indoor residual spraying was done periodically, several barriers were identified.

### Conclusion

Increased awareness of malaria prevention and treatment among people living in malaria risk areas is important for the National Malaria Program in its quest for malaria elimination in Nepal.

**Funding:** The principal investigator received funding support from the Research Training Program scholarship at Curtin University. However, the funders had no role in study design, data collection and analysis, decision to publish, or preparation of the manuscript.

**Competing interests:** The authors have declared that no competing interests exist.

## Introduction

Globally malaria contributes 409,000 deaths annually [1, 2]. The number of malaria cases in Nepal decreased from 11,000 in 2000 to 1,065 in 2019 and a low Annual Parasitic Incidence of 0.09 per 1,000 population at risk [3]. Among these 1065 malaria cases, 440 were indigenous (locally transmitted) and 625 were imported cases mostly from neighbouring India and countries in Africa [3]. Plasmodium vivax constituted 94.6% of the total cases in the country, whilst the remaining 5.4% were Plasmodium falciparum cases [3]. The decrease in cases is attributed to scaling up malaria prevention interventions including; the free distribution of long-lasting insecticidal nets (LLINs), periodic indoor residual spraying (IRS) of insecticides in high-risk areas, improved surveillance and active case detection, and the use of Artemisinin based Combination Therapy (ACT) for treatment of *P. falciparum* malaria, and support from external development partners such as the Global Fund [3, 4]. Nepal is well on course to reach elimination by 2025 as outlined in its National Malaria Strategic Plan 2014–2025 [3, 5].

Nepal is divided into seven provinces, of which two, Karnali and Sudurpaschim, contributed more than 70% of the country's malaria burden in 2019 [3]. Geographically and topographically the country is divided into three distinct areas: the Terai plains in the south bordering India; middle hills; and upper mountains in the north adjoining Tibet [6]. The Terai plains constitute a large area of national parks and forests, which experience a subtropical climate, year-round cultivation, and constant cross border movement of people from Nepal to India through open borders [6]. The valleys between the middle hills and the upper mountains are often referred to as the upper river valleys (URV). The URV corridor recorded four separate outbreaks in six villages across five districts between 2017 and 2019 [7, 8]. In 2019, the URV contributed to 28% of the overall malaria cases in Nepal [7, 8]. Between 2017 and 2019, the largest numbers of cases (289, mostly indigenous *Plasmodium vivax* cases) were reported from Rigga, a village in Khatyad Rural Municipality (KRM), located in the URV of Karnali province [3, 8, 9]. The URV corridor is remote and has limited road access, creating significant challenges for preventing and controlling malaria transmission.

Temperature, humidity and rainfall are important environmental factors linked to malaria transmission [10]. Summer temperatures across the URV reach as high as 40 degrees Celsius and the rain during the monsoon season creates water reservoirs that act as mosquito breeding sites [11]. Studies have shown that malaria vectors can survive up to 3,000 meters above sea level [10]. The ability of vectors to survive and breed at higher altitudes is a cause for concern when it comes to preventing the transmission of malaria in the remote URV corridor.

Malaria is often considered a disease of the poor and has been associated with communities [12, 13]. The majority of people living in remote villages in the URV rely on traditional small-scale farming and have limited household income [14]. Families of lower socioeconomic status (SES) are less likely to seek health care early, leading to delayed malaria diagnosis and increased symptom severity [14, 15]. Delays in malaria treatment often lead to increased transmission and outbreaks. Moreover, a scarcity of funds influences household priorities, and purchasing bed nets and other prevention measures is not prioritized [13]. Furthermore, limited sources of income coupled with modest employment opportunities lead to villagers migrating to malaria endemic urban Nepal or neighbouring India, potentially exposing them to malaria [6, 16]. The seasonal back and forth movement of potentially infected migrants from high risk malaria areas, coupled with presence of the vectors locally can trigger malaria transmission in the community [16].

Nepal's National Malaria Elimination Program (NMEP) distributes LLINs through mass campaigns in endemic areas every year [17]. Regular use of LLINs is dependent on various factors such as awareness of their importance, temperature (use can decline with hot weather),

distribution of free LLINs from the NMEP and availability of hanging space in small rooms [18]. Additionally, periodic IRS in endemic areas of the country is carried out during pre and post-monsoon season by the NMEP [3, 5]. IRS reduces malaria incidence in highly endemic areas and is an important strategy for preventing malaria transmission [19, 20]. Local epidemiological data and vector behaviour studies are used to direct teams during an IRS campaign. Community acceptance is paramount to the success of any IRS campaign and several rounds of IRS within short time frames may cause fatigue and irritation among household members, leading to households refusing to allow the activity to be undertaken [20]. Given the barriers to the success of the malaria control interventions, this study aimed to explore the knowledge and attitudes about behaviour towards malaria prevention, and explain factors that might influence treatment among people living in two malaria-endemic wards (smallest administrative unit) of KRM.

## Materials and methods

This qualitative research used a phenomenological approach to explore the real-life experiences of the Nepalese people living in the KRM [21]. The study was approved by the Nepal Health Research Council (ERB 632/2020, Ref. No. 1287) and Curtin University's Human Research Ethics Committee number HRE2020-0701. The design and the results of the study followed the Consolidated Criteria for Reporting Qualitative Research (COREQ), with respect to: 1) the research team and reflexivity; 2) study design; and 3) reporting the results [22].

### Research team

The research team comprised of six members located in Australia and Nepal. A Nepalese-born doctoral student enrolled in an Australian university, experienced in qualitative and quantitative research and community facilitation (KA); three university based academics with expertise in malaria epidemiology (AC), qualitative research and health promotion (JEL and JJ); and local research assistants (RAs) with a public health background and currently working for the national malaria program at the study site.

### Study setting

The study was conducted in a remote URV high malaria risk village of KRM located in Karnali Province, Nepal [23]. The sites were selected based on past episodes of malaria transmission, with the wards classified as high-risk wards by the NMEP in their 2019 microstratification exerciseMajority of the population in this area rely on subsistence agriculture or seasonal labor migration for income and have a lower socioeconomic status [3]. The nearest road access is an hour by foot. The roads are not accessible during the monsoon season (June-August) due to rain, or during winter (January and February) due to snowfall. The closest public health facilities, the health posts (HPs), Ama HP and Hyanglu HP, are half an hour walk away, whilst a Primary Health Centre (PHC), Ratapani PHC, is a further two hour walk by foot. A private clinic with a pharmacy, locally referred to as a 'medical', is an hour walk from the villages.

### Participant eligibility and recruitment

Twenty-five participants were purposively selected, including males (n = 10) and females (n = 15) between the ages of 15 to 72 years. The inclusion of elderly participants aged 60 years and above was to explore their experiences of malaria, both past and present, along with their roles in health related decision-making within extended families [24]. A younger age group was also recruited to explore the influence of cultural practices among adolescents (n = 2)

whilst female community health volunteers were included (n = 4) to elaborate the health care seeking behaviour of the community. Participants were selected using a criterion-based sampling method, a criteria adopted to select a variety (farmers, local leaders, migrants, school teachers, and students) of participants, including males, females and people from varied age groups [13, 25]. Based on these criteria, a mix of potential participants were identified by the principal researcher from different cohorts that included priests, health workers, past malaria patients, teachers and students to generate rich information. The RAs supported in recruitment of these participants at the study site and arranged the necessary logistics for conducting the virtual interviews.

## Qualitative data collection

**Interview guide development.** A semi-structured interview guide was developed based on similar research [17] and the researchers' experience exploring the topic of malaria. In-depth interviews are used in public health research to explore the perspectives, experiences and thought processes of individuals and to understand how these influence behaviour choices [26]. Piloting of the interview guide was conducted (n = 3) to ensure there was no ambiguity in the questions. The comprehension of malaria specific terminology were assessed and questions were modified or excluded based on the feedback from the pilot stage.

**Data collection.** Virtual in-depth interviews were conducted between November and December 2020 (post monsoon) by the lead author (KA) with local support from Research Assistants. Due to COVID 19 travel restrictions, the interviews were conducted virtually using video conferencing. Whilst sixteen participants were interviewed in their own house, remaining nine including health workers were interviewed in a meeting hall of the nearby health post. Participants provided consent, both oral and written, before participating in the study and the written consent were collected by the RAs. Personal identifiers were neither recorded nor transcribed to maintain anonymity. Permission was also obtained to audio record the one-on-one interviews. All interviews were conducted in Nepali. Interviews were between 25 to 35 minutes in length. Anonymity was ensured by using pseudonyms and removing any identifiers including names and data were stored securely.

**Data analysis.** Audio recordings of the interviews were transcribed verbatim in Nepali and later translated into English by the lead researcher (KA). Inductive thematic analysis was used to identify and report the themes generated from the data using Braun and Clarke's method: familiarizing with the data; generating initial codes; searching for appropriate themes; reviewing the identified themes and defining/ naming the themes [27]. Transcripts were read thoroughly twice and coded using NVivo software version 12 (QSR International Pty Ltd) by the lead researcher KA. Codes were created using an open coding process to explore all new ideas and concepts. Although the initial coding concentrated on the language and descriptions used by the participants, the information from the generated codes helped to identify the major concerns. The generated codes were organized and sorted based on similarities and clustered into core themes. Literature on malaria prevention and treatment was read concurrently during the data analysis to help understand and develop the generated themes. Themes were also identified independently by JEL and JJ and consensus was reached on the final themes. A concept mapping process was used to explain the relationships and associations between the generated themes based on the focus of the study [27, 28].

## Results

The results of the interviews (n = 25) including illustrative quotes of the participants are presented. The participants included farmers (n = 13), priest/ traditional healers (n = 2), female

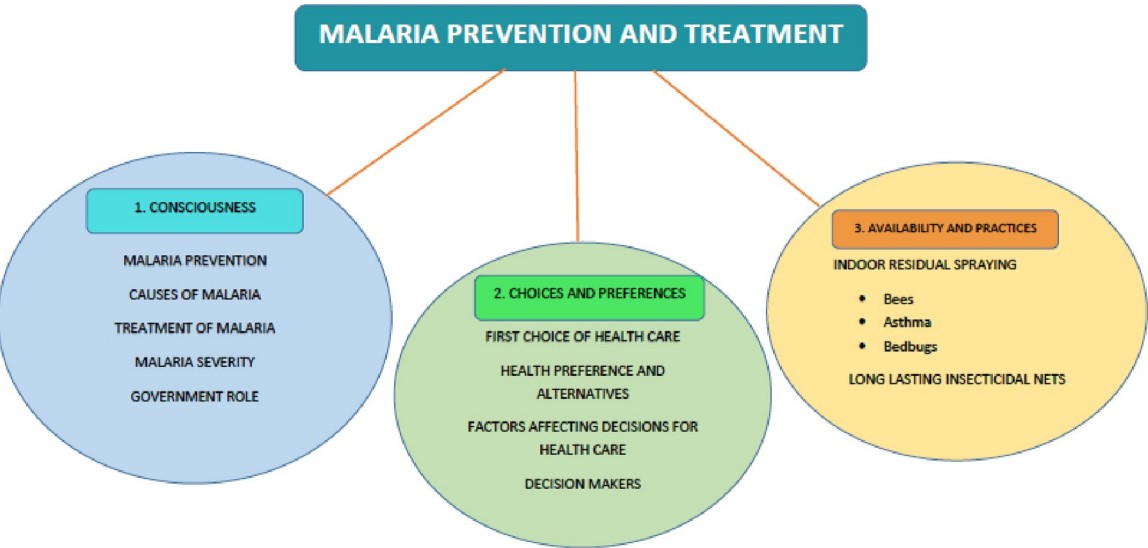

**Fig 1. Concept map of identified themes.**

community health volunteers (n = 2), school teachers (n = 2), students (n = 2), self-employed/service (n = 2) and local leaders (n = 2). The central concept '*malaria prevention and treatment choices*' and the associated knowledge, attitudes and behaviours are discussed. Three themes were identified: 1) consciousness; 2) choices and preferences; and 3) availability and practices (Fig 1).

## Consciousness

More than half of the participants (n = 17) correctly identified "*Gamgadas*" (the local name for mosquitoes) as the cause of malaria, however, others were unclear about the actual cause. Two participants felt wearing thin clothing facilitated mosquito bites. A participant suggested that the environment around the house was related to an increase in mosquitoes that cause malaria "*When there are ditches or potholes, dirt around the houses, mosquitoes are produced in the standing water in the ditches*" (male, farmer). The remaining participants, however, had minimal ideas on malaria transmission. While some participants blamed the unclean water during the monsoon season for causing malaria, others associated it with undernutrition, consumption of stale food and sun (hot weather). A high school student shared her malaria knowledge "*It [malaria] happens if we eat unhealthy and rotten foods, if we don't wash our hands, if we drink dirty water, if we don't wash our clothes and if we don't wear thick clothes*" (female). With regards to the severity of malaria, three of the participants thought that malaria would make them physically weak, while eight of the participants thought it could result in death.

Most of the participants (n = 19) knew that they could get anti-malarial medicine from the public HPs whilst fourteen participants knew they had to take two different types of tablets for a period of 14 to 15 days, a common duration required to treat *Plasmodium vivax* infection. Five participants however reported incidents of non-adherence to the treatment protocol in their family and neighbours. "*They [villagers] take medicines for 4–5 days and feel better, and then they leave the medicine as it is. Some people did not have it [full medication course] due to shivering and thought that a reaction had happened. We [community health volunteers] and the health workers all went there [patient's house], convinced them, and made them take the medicines.*" (female, FCHV). She also recalled a relapsed malaria case that she had attended "*She*

*[patient] threw away the medicines after taking only half . . .. She had malaria again when she was pregnant".*

Altogether thirteen participants mentioned that the government had conducted programs including distribution of LLINs, IRS and community testing in the village. The remaining participants were either unsure or unaware of the role the government had played in the prevention of malaria in their village whilst three participants credited the support provided by external development partners (Non Governmental Organisations and International Non Governmental Organisations) for their role in malaria prevention and treatment.

## Choices and preferences

**First choice of health care.** All the participants said that their first choice of health care was the public HP nearby. Participants preferred the public health facilities because of their proximity, free service and provision of medications *"At first we will go to the health post. If the treatment is not done there [public health post] then we will go to the private medical [clinic]. Poor people can get free malaria treatment at the health post"* (female, farmer). However nine participants identified limitations of the public HPs including inadequate resources and limited hours of outpatient services due to which they were compelled to seek service from the nearest private medical clinics. One participant commented *"When the required medicines are not available in the health post, we go to the medical [private clinics]. We do check-up in the health post and get the medicine from the medical"* (male, farmer).

For tertiary (advanced) hospital care including treatment of severe malaria, the participants explained that they had to either travel to Nepalgunj (the nearest city) or Kathmandu (the capital). Participants noted that in the past villagers would go to Nepalgunj for malaria treatment *"Two-three years back, most of the people went to Nepalgunj for treatment of malaria. The main city near to us is Nepalgunj" (female, farmer).* Travel to Nepalgunj and Kathmandu was an expensive, arduous journey taking several days *"It takes a lot of time sir to go downwards [Nepalgunj]. . . it takes 4–5 days"* (male, farmer). Although participants could fly to Nepalgunj from Kolti in Bajura, a four-hour walk from the villages, the flights were limited *"A lot of times the planes do not fly. They [airlines] say the weather is not good. . .. as passengers are very few. . . there are no people coming up here [Bajura] from down there [Nepalgunj]"* (male, photographer).

Traditional management of malaria was described by participants as visits to traditional healers known as *'Dhamis'* or *'Jhakris'*. However, the use of traditional healers did not seem common. The participants reported that they would not seek care from *Dhamis* for malaria now *"I don't believe in Dhami-Jhankri sir. I am also one of them sir but I don't believe"* (male, priest). Nonetheless, the Dhamis' advice was still sought for other illnesses including those related to faith *"When we suspect someone being possessed by Lagubhagu [evil spirits] we go and visit the Dhami Jhakri, and show it to them"* (female, farmer).

**Factors influencing health decisions.** The participants highlighted that distance to health care facilities, cost of treatment and the decision making authority of the head of the family, were factors that influenced decisions regarding health care choices and timely treatment. The participants shared the difficulties in accessing health care for different age cohorts and the severely ill due to the remoteness and distance to commute on foot to the nearest health facility. *"We get dizziness, leg pain, bone pain while going there [health post] and we become tired after returning"* (female, mothers group member). Another participant further described the difficulties surrounding several age groups and conditions *"there are some people like old people, small children . . .for us, there is no problem. . . let's say if there is delivery situation [childbirth] or sick elderly people, they have to be carried to the health post"* (male, photographer).

The cost of treatment influenced health-related decisions often presenting as barriers. A participant highlighted the cost of the journey to get a family member treated for malaria after being referred to a public tertiary hospital in Nepalgunj *"We took him in the plane, it cost NRs 5,000 [USD 50] and thereafter it took more than NRs. 60,000 to 80,000[USD 550–750]"* (female, farmer/ housewife). Decision makers within a family were found to play a key role in health decision making and were influential in determining where the treatment is sought such as traditional versus public health facilities or private clinics. According to seventeen participants, the decision makers regarding health matters in the family were often the family head, mostly males (father/father-in-law/husband). However, among families where males were not around, the wives (n = 3) were the key decision makers. *"My father is not alive. My mother takes it [the decisions]"* (male, farmer).

## Availability and practices

**Use of long lasting insecticidal nets.** The provision and subsequent use of LLINs from the Epidemiology and Disease Control Division (EDCD) in 2018 were explored. All the participants acknowledged receiving LLINS whilst most of them (n = 22) revealed they only used them for three to six months and not during the winter (December to February) as the mosquitos were sparse. However, three participants shared that they used the nets all year to protect themselves from other insects *"There are 'Chadchu' [bedbugs] all the time, so we keep using it both in the rainy and winter season"* (female, farmer).

Some participants (n = 3) shared that the nets distributed in 2018 were torn; the causes were frequent washing and use of detergents, including baking soda. A participant highlighted why it was necessary to frequently wash the nets *"In the house, there is only one kitchen and same place for sleeping, it [LLIN] becomes black"* (male, photographer). An elderly female explained the reason for using a detergent or baking soda while washing her nets, and how it affected the LLINs *"When we kill the mosquitoes and the 'Chadchu' [bed bugs] the nets get stained with blood which would dry up. So when I washed it in the hot water with soda, again and again, to get the blood out, it tore out"* (farmer). Ten participants mentioned that washed nets were dried in a shaded area as recommended by the manufacturer.

**Indoor residual spraying.** All the participants revealed that IRS was done in their houses every year. They shared that the spraying killed all the insects including mosquitoes and cockroaches in the households and believed it had contributed to the decrease of malaria cases in the village. However, fifteen participants reported experiencing several negative consequences of IRS, which caused some villagers to resist or refuse insecticide spraying in their houses. Altogether eleven participants shared their experiences of increased bed bugs after the spraying episodes *"After spraying, the places where spraying was done, bed bugs were seen there. I think bed bugs were produced after that"* (female, primary school teacher). One participant stressed how the IRS had disrupted the whole animal food chain that led to an increase in bed bugs. He presented his theory, *"There are many bed bugs even in the winter and don't let us sleep. Earlier, they [bed bugs] would be eaten by the cockroaches and lizards, but after the spraying, all cockroaches died and lizards went away so nothing to kill them [bed bugs]. My neighbour got cockroaches from her maternal home this year and reared them and is saying that the bed bugs are so less now. Maybe I will go and ask her for some [cockroaches] (laugh)"* (male, serviceman).

Nine participants reported the consequences of IRS on the bee-farming family business. The people in the villages traditionally rear bees for honey inside their houses. Participants stated that their bees died or fled due to the insecticide (IRS) *"Some villagers said that their 'Mauras' [bees] would die and they have to bear the loss, they don't let them spray [insecticides] in their houses"* (male, priest). For these villagers, the need to maintain their bees and source of

livelihood, outweighed the potential health risks posed to them and the community from a lack of spraying. Lastly, five participants held negative views towards IRS due to concerns about the effect on existing health conditions, especially respiratory conditions like asthma. *"They [villagers] requested not to spray as they had asthma patients and bees at their home"* (female, teacher). Several participants (n = 3) commended the FCHVs and health personnel for their efforts in advocating and motivating the resisting villagers to support the preventive IRS activities.

## Discussion and recommendations

This research explored the knowledge, attitudes and behaviour of the people of KRM with respect to malaria prevention and treatment. Our results found that many participants had a high level of current knowledge of malaria prevention and treatment. Factors such as cost, distance to the health facility and decision making authority among elders influenced health seeking behaviour in the community. This study will add to the limited literature on malaria in Nepal and inform the NMEP in formulating strategies to remove community-level barriers for malaria prevention and treatment.

The interplay of social and economic factors contribute to the complexity of malaria prevention and treatment in rural settings. This study found that, whilst half of the participants correctly identified the causes of malaria, several participants were unaware of the causes of malaria. Similar to our findings, Togbay et al. in Bhutan found participants perceived hard work in adverse weather conditions (heat and rain), cooking with firewood, dirty and unhygienic surroundings in and around the house, dirty water, and people sleeping outdoors during the harvesting as causes of malaria [29]. Described as a disease of the poor, knowledge related to malaria disease and transmission is associated with the SES [12, 30]. Yadav et al. in Rajasthan India, reported that 90% of their respondents from lower SES communities were not aware that mosquito bites caused malaria compared to 36% being unaware among those of higher SES [30]. Of interest, the participants knew that untreated malaria could make people severely ill, weak and even lead to death. As the majority of the population in KRM belongs to the lower SES group, future health promotion strategies targeted at awareness raising amongst lower SES groups on causes and prevention of malaria would be worthwhile.

Treatment adherence to anti-malarial drugs was found to be an issue. Participants were aware of anti-malarial medication and knew that it was provided free of cost from the public health facilities, however, our participants reported non-compliance with the treatment protocol after the third day as they were asymptomatic. The Nepal national malaria treatment protocol (NMTP) recommends two drugs for the treatment of *Plasmodium vivax;* a three-day Chloroquine and a two-week Primaquine course [31]. Interestingly, compliance issues exist in countries with the shorter seven-day Primaquine regimen, therefore it is a challenge for the NMEP to ensure Primaquine adherence for fourteen days, raising concerns of possible future relapses [32, 33]. Grietens et al. found that only half of their study participants completed a seven-day Primaquine course, with one-fourth experiencing relapses as a result of non-adherence [32]. Reasons for non-compliance included easing of symptoms within the first three days, bad taste of the medicine, loss of appetite, and allergies [32]. A multi-country randomized control trial revealed similar recurrence rates in a higher dose seven-day (1·0 mg/kg per day; 7·0 mg/kg total dose) and lower dose fourteen-day Primaquine regimens (0·5 mg/kg per day; 7 mg/kg total dose) [33]. This finding could provide the NMEP evidence to revise the existing NMTP. Nonetheless, there is a need to improve awareness in the population regarding the possibility of relapse due to Primaquine non-adherence. Directly observed therapy (DOT) of Primaquine through health workers, especially FCHVs, could be a viable way to increase compliance [34].

The participants were not aware of the role of the government in malaria prevention and management, which reinforces the need to increase the visibility of the NMEP at the grassroots level. In an effort to meet the program goals, the NMEP should work collaboratively with the local municipality. Therefore, a combination of top-down and bottom-up approaches when designing, implementing and evaluating malaria prevention activities will be one way to approach malaria prevention and control [35, 36]. Mlozi et al. in their study in Tanzania found that community engagement in malaria control interventions empowers people at risk, improves disease-specific knowledge and increases acceptability amongst targeted communities [36]. The involvement of experts from the government and health sector along with community members in all phases of the intervention will decrease the gap between the perceived needs and the actual needs of a population, thereby developing local ownership of the activities. This will make the population at risk aware of the national goals and ensure long-term sustainability.

This study found participants preferred public health facilities for malaria diagnosis and treatment compared with private and traditional healers. Malaria treatment is also free in these public health facilities across Nepal [7, 8]. Of interest the villagers still sought health care form *Dhami's* and *Jhakri's*, for diseases other than malaria. The participants including a *Dhami* stressed the need to take medication from the HP to cure malaria. This is a positive behaviour change, suggesting an acceptance of western-style health care service providers. Our finding contrasted with other countries where traditional healers still play a role in malaria diagnosis and treatment. Among rural indigenous communities in the Philippines, Matsumoto et al. [37] found a strong inclination of people to seek care for malaria treatment from the *Albular-yo's* (traditional healers). Strong cultural and traditional influences coupled with doubts on the treatment provided by health professionals were the factors influencing choices [37]. The NMEP should explore the reasons for this change in attitude on disregarding traditional healers for management of malaria and replicate it in other risk areas where it is still practiced.

Our study revealed certain limitations of public health facilities that compelled patients to seek service from private providers. Similar to a study undertaken in Bhutan, limited hours of outpatient services, and limited availability of resources including medication and diagnostic kits throughout the year were found to divert patients to private clinics [29]. Private clinics play an important role in bridging health care gaps in remote settings. However, the quality of diagnosis in treatment of malaria cases in these clinics is questionable, due to the weak monitoring system and lack of commercially available Primaquine in Nepal [12]. A multi-country study on the private sector role in malaria case management illustrated that informal providers (outlets run by individuals with minimal or no training) were responsible for up to 77% of health care interactions in Bangladesh, especially among in the rural poor [38]. Most private clinics in remote villages are not registered and cannot report to the national Health Management Information System (HMIS), resulting in missing cases in the surveillance system. It is, therefore, important for the NMEP to engage private sector providers in malaria risk areas in surveillance activities whilst updating them on the NMTP to ensure correct treatment [38]. Our study showed that referral to tertiary centres added financial burden to families, whilst the difficulties faced to travel further amplifies the need to ensure year-round availability of malaria diagnostic and treatment at facilities within the vicinity of the malaria-endemic wards. Such accessibility and availability would be crucial to prevent any local transmission or focal outbreaks. Outbreaks in such remote areas would be very costly, and difficult to contain promptly, whilst management of possible severe or complicated malaria (in case of Plasmodium falciparum) in the absence of tertiary care facilities in the locality would increase mortality risks.

Participants identified a range of factors that prevent timely access to treatment, namely cost, underlying physical condition and the existing patriarchal social structure. Cost of treatment is a barrier to accessing health care worldwide; this was evident in our results where it influenced health decisions among the participants [12]. In addition, Nepal does not have a universal health insurance system, and all health care costs are out-of-pocket (OOP), often delaying health care seeking among the poor [12, 30, 39]. Therefore the option for free malaria treatment is a step towards equitable and early care, thereby preventing severe malaria [5]. Physical condition such as weakness due to illness or old age of the participants was associated with altered health care seeking behaviour. In remote areas with limited or no transport infrastructure, a long commute by foot was reported to prevent early presentation for treatment [12, 40]. Such delays in malaria treatment can result in local transmission and focal outbreaks. In the Mekong region, active case detection using mobile health workers has ensured that the vulnerable and physically weak are tested and treated in the community mitigating travel concerns and preventing outbreaks [35, 41]. This strategy is very effective for rural, hard-to-reach areas that are often underserved by the national malaria programs due to limited access, cost and logistic difficulties [35]. Nepal is known as a patriarchal society, and we found that key decisions in the family were made by the elders and household heads (mostly males). A household in KRM has an average of six family members, a family size suggestive of large extended families [42]. Financial and health-related decision making authority in extended families lies with the parents or in-laws, leaving the young and especially the daughters-in-law vulnerable [43]. The NMEP should aim to engage and involve the elderly in social awareness activities to bring about desired changes in behaviour and decision making regarding prevention and treatment of malaria. A community advisory board comprising of the elderly to lead and implement activities was found to be widely accepted in the community and successful in malaria prevention in Laos [44]. Such a model could be replicated in highly endemic areas of Nepal to promote local level leadership, and involvement in implementing the programs and activities.

Our study identified certain barriers to preventive measures such as LLIN use and IRS. The majority of our participants used LLINs for three to six months each year and not during the winter. Sahu et al. in Orissa, India, noted a similar behaviour where 24 to 26% of the participants used LLINs seasonally, mostly during the rainy season [45]. Similar behaviour regarding periodic use of LLINs was reported in other studies such as the one conducted in Bhutan [29]. Similar to our findings, the reasons for underutilization of the LLINs in India and Bhutan included a lack of availability of LLINs, early attrition of the nets, reliance on other protective measures such as fire, smoke or fans, and low mosquito density in the houses during winter [29, 46]. LLINs are not available commercially in Nepal; a factor in the sparse use of nets among the participants. Therefore, the NMEP should coordinate with the local municipalities with a high risk of malaria to replace the LLINs periodically. The NMEP could also collaborate with private sector providers to make insecticide treated nets available and affordable commercially, allowing access for those that require a replacement in case of early attrition of the distributed nets [47]. Additionally, normal nets can still act as physical barriers to mosquitoes and other insects including bedbugs, therefore in the absence of LLINs, the use of these nets should be promoted. Interestingly, our participants used detergents and baking soda to wash LLINs to remove dried blood stains and discoloration caused by cooking using firewood in the kitchen. Even Sahu et al. found that 79% of their participants used a detergent or soap to wash LLINs in India [45]. Using coloured LLINs to minimize dirt visibility and making users aware of how detergents and soda can decrease the efficacy of the LLINS and durability, could facilitate behaviour change.

The participants highlighted the unintended negative consequences from IRS such as loss of bees, an increase in bedbugs and allergic reactions in the elderly. Negative consequences could promote community resistance and create further barriers to IRS roll out. Going forward it would be worthwhile to explore the possibility of moving the beehives outdoors to the fields so that they are not affected by the IRS. Locally available plants such as *Citronella* and *Eucalyptus* have shown repellent characteristics against both mosquitoes and bed bugs with a mortality rate of over 70% over a 24 hours exposure [48]. Such locally available plant derivatives could complement the IRS in addressing concerns around bed bugs. Interestingly, community resistance to IRS in other countries slightly differed from our findings. Barriers identified by respondents in an Indian study included people believing it increased rodents, left a bad smell, was tiring to shift household goods during spraying, and feared poisoning food, drinking water and domestic animals [30]. In Mozambique, Macago et al. identified additional factors such as disagreement over the selection of spray-men, negative experiences from previous IRS events and political factors such as the difference of opinions between the local leaders of different political parties on key decisions creating community resistance to IRS among the educated population [49]. The NMEP should take into account the negative experiences described by our participants and make the target population aware of the benefits of IRS while removing any misconceptions.

## Strengths and limitations of the study

This study captures the perspectives of malaria prevention and treatment of the local community of Khatyad RM of Nepal. The study was conducted by a Nepalese researcher in the native language, which allowed the participants to fully express their feelings and views. The questions were pretested and the study was guided by the COREQ [22]. The inclusion of participants between 15 to 72 years of age allowed us to explore malaria prevention and treatment practices and changes across generations. Due to COVID -19 restrictions, the interviews were conducted virtually, which at times compromised the flow of the interviews due to limited internet services. However, the communication and conversation during the interviews was very similar to conducting face-to-face interviews. As information was collected only using virtual interviews, triangulation could not be done, a limitation of the study. Nonetheless, to our knowledge, this is the first study conducted in the rural malaria endemic areas of Nepal to explore the knowledge, attitudes and behaviour of people relating to malaria prevention and treatment.

## Conclusion

In remote, malaria-endemic villages in Nepal, the choice of malaria treatment and healthcare is dependent on various factors such as cost, distance to travel and the knowledge and attitudes of the decision makers in the family. Furthermore, despite public HPs being the first option for malaria treatment and health care, a lack of resources and timely supply of diagnostic and treatment services are some of the limitations faced by people in rural areas. Despite the NMEP providing free LLINs and spraying IRS periodically, negative experiences coupled with a lack of awareness have resulted in the intervention not eliciting the required behaviour change nor full coverage. The NMEP needs to take note of these barriers and plan and implement strategies to overcome the barriers and enable change them. This would ensure maximum benefit from the preventive measures being implemented to prevent malaria transmission in rural communities and bring about sustained behaviour change.

## Supporting information

**S1 File. Domains of enquiry.**
(PDF)

## Acknowledgments

We would like to acknowledge the support of Mr. Mukunda Karki, Mr. Rohit Sah, Dr. Madan Koirala, and Ms. Tamanna Neupane for their support in conducting the study at the field level. We also would like to thank, the participants for their time and the officials of KRM for allowing us to conduct our research in their municipality.

## Author Contributions

**Conceptualization:** Kiran Raj Awasthi, Jonine Jancey, Justine E. Leavy.

**Data curation:** Kiran Raj Awasthi.

**Formal analysis:** Kiran Raj Awasthi.

**Investigation:** Kiran Raj Awasthi.

**Methodology:** Kiran Raj Awasthi.

**Project administration:** Kiran Raj Awasthi.

**Software:** Kiran Raj Awasthi.

**Supervision:** Jonine Jancey, Archie C. A. Clements, Justine E. Leavy.

**Validation:** Kiran Raj Awasthi, Jonine Jancey, Justine E. Leavy.

**Writing – original draft:** Kiran Raj Awasthi.

**Writing – review & editing:** Kiran Raj Awasthi, Jonine Jancey, Archie C. A. Clements, Justine E. Leavy.

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
