## [Decision Letter · Decision Letter 0]

13 Oct 2021

PONE-D-21-28469A Qualitative Study of Knowledge, Attitudes and Perceptions towards Malaria

Prevention among People Living in Khatyad Rural Municipality of Mugu, NepalPLOS ONE

Dear Dr. Awasthi,

Thank you for submitting your manuscript to PLOS ONE. After careful consideration, we feel that it has merit but does not fully meet PLOS ONE’s publication criteria as it currently stands. Therefore, we invite you to submit a revised version of the manuscript that addresses the points raised during the review process.

Kindly address all comments from both reviewers, specifically add to the limitations section as suggested by reviewer 1 and re-analyze your findings in the light of the identified limitations. Reviewer 2 has pointed out that presentation of quantitative data through the manuscript is incomplete and I would kindly ask you to address this issue accordingly.

We look forward to receiving your revised manuscript.

Kind regards,

Benedikt Ley

Academic Editor

PLOS ONE

Journal Requirements:

2. Please include additional information regarding the survey or interview guide used in the study and ensure that you have provided sufficient details that others could replicate the analyses. For instance, if you developed a survey guide as part of this study and it is not under a copyright more restrictive than CC-BY, please include a copy, in both the original language and English, as Supporting Information.

4. We note that Figure 2 in your submission contain copyrighted images. All PLOS content is published under the Creative Commons Attribution License (CC BY 4.0), which means that the manuscript, images, and Supporting Information files will be freely available online, and any third party is permitted to access, download, copy, distribute, and use these materials in any way, even commercially, with proper attribution. For more information, see our copyright guidelines: http://journals.plos.org/plosone/s/licenses-and-copyright.

Additional Editor Comments:

Kindly add to the limitations section within the discussion as suggested by reviewer 1, and address all comments of both reviewers.

Reviewers' comments:

Reviewer's Responses to Questions

**Comments to the Author**

1. Is the manuscript technically sound, and do the data support the conclusions?

Reviewer #1: Partly

Reviewer #2: Yes

2. Has the statistical analysis been performed appropriately and rigorously? 

Reviewer #1: N/A

Reviewer #2: Yes

3. Have the authors made all data underlying the findings in their manuscript fully available?

Reviewer #1: No

Reviewer #2: No

4. Is the manuscript presented in an intelligible fashion and written in standard English?

Reviewer #1: Yes

Reviewer #2: Yes

5. Review Comments to the Author

Reviewer #1: This paper presented a qualitative study of people living in the remote village of Nepal that is considered to have high malaria risk by the NMEP, exploring the knowledge, attitudes, and behavior towards malaria prevention. Understanding community perception is key to disease control and prevention, particularly among communities living in remote villages where contact with health workers may be scarce. However, this paper has some methodological weakness and needs additional information on malaria epidemiology and social context.

Major comments

This paper presented findings from only one method and source of data collection and did not mention triangulation to test the validity of the findings. While the pandemic might restrict certain data collection methods such as observation, other strategies to test validity through the convergence of information from different sources could be sought.

Minor comments

The lack of information on malaria incidence in the study site made it difficult to interpret the findings on knowledge and awareness into context.

As the researchers were not present in the study site, the role of research assistants also needs to be explained more to reflect the sample selection and data collection process.

Section by section review

1. Introduction

Paragraph 1 (lines 45-52):

It would be helpful for the context of the research to have epidemiological measures both nationally and in the study site for the readers to get an idea of how big of a burden is malaria in Nepal and in the study site. The first paragraph of the introduction had numbers of malaria cases, but without any denominators it is difficult to put the information in context. I would suggest adding extra information such as annual malaria incidence. Having information on the proportion of P. vivax malaria will also make the result and discussion on treatment adherence more relevant.

2. Methods

Research team (lines 105-110):

Because the researchers were not present in the study site during data collection, it is important to describe in more detail about the local research assistants, who they are, what were exactly their role in the study, and what were their relationship with the study participants.

Participant eligibility and recruitment (lines 121-127):

I assume this research was done during the covid-19 pandemic so I expect some restrictions in contact between people. If this was the case at the study site, I think it is relevant to explain the method of approach and any non-participation.

Data collection (lines 137-145):

In the discussion section (line 453-454) it was mentioned that the majority of interviews were done virtually. How many were done in person and how many were done virtually?

Where were the zoom interviews held in the study site? Is it at a clinic? At participants’ own houses? Or elsewhere?

3. Results

In some of the quotes the participants mentioned their role as community health volunteers. It would be good if the results section started with a description of such characteristics. How many of the participants were community health volunteers? What about the other participants? What are their roles and positions in the society?

There is a paragraph in the discussion section about treatment adherence. Is there any data that can be added in the results section that might suggest malaria relapse or recurrence?

4. Discussion

The lack of test of validity of the findings due to using only one type of data collection methods with no triangulation should be mentioned as one of the limitations.

5. Title and abstract

As not all readers would be familiar with places in Nepal, I personally think it might benefit the paper if the title and/or abstract mentions more descriptive words such as “remote” and/or “upper river valley.”

Reviewer #2: Review of the paper “A Qualitative Study of Knowledge, Attitudes and Perceptions towards Malaria Prevention among People Living in Khatyad Rural Municipality of Mugu, Nepal” by Awasthi et al.,

Please find below my comments

Methods

Line 121 : Participant eligibility and recruitment : for in-depth interviews, participants are generally people who interact with many people in the community on a daily basis e.g. -head of districts, religious leaders, youth or women leaders etc.- as they are well informed on the population KAP… Pleased clarify if those people were included as participants. Also provide more details on the criteria used for the selection of participants.

Line 130 “A semi-structured interview guide was developed based on similar research [17] and the researchers’ experience exploring the topic of malaria” Can the author add as additional file the questionnaire or the guide used for the interview.

Was survey done during the rainy or dry season? Please clarify as this might affect perception or risk/attitudes toward malaria prevention and treatment.

Results

The author need to be more precise on the number who responded over the number interviewed.

For instance Line 183 “Most of the participants knew that they could get anti-malarial medicine from…” How many please give the number

Another Line 190 “Only half of the participants mentioned that the government had conducted…” They interviewed 25 persons what do they consider as half is it 12 or 13 people please clarify. There are many places in the document where similar terms are used they need to check the whole result section and provide full details on the number of respondent.

See again below

Line 264 : “The majority of the participants (n=10) informed that washed nets were dried in a shaded area as recommended by the manufacturer” is 10 out of 25 interviewed represents the majority ?

also see line 271 “However, most of the participants reported experiencing several negative consequences of….” Wow many please give the precise number

Line 273 “Nearly half of the participants (n=11) shared their experiences of increased bed bugs after…”. Please check and correct accordingly.

Line 296 : I don’t see the need of adding this figure

6. PLOS authors have the option to publish the peer review history of their article (what does this mean?). If published, this will include your full peer review and any attached files.

Reviewer #1: No

Reviewer #2: No

---

## [Author Response · Author response to Decision Letter 0]

18 Dec 2021

Response to reviewers

Manuscript title: A Qualitative Study of Knowledge, Attitude and Perceptions towards Malaria Prevention among People Living in Rural Upper River Valleys of Nepal

Dear reviewers, thank you for your comments and suggestions which have been addressed and colour coded in the manuscripts for your kind perusal. We would also like to acknowledge the effort and time that you have provided to go through our manuscript. 

Reviewer 1

Comment 1

This paper presented findings from only one method and source of data collection and did not mention triangulation to test the validity of the findings. While the pandemic might restrict certain data collection methods such as observation, other strategies to test validity through the convergence of information from different sources could be sought.

Response 1

Thank you for your suggestion. This has been now highlighted as one of the limitations of the study. Please refer to Page 22 Line 475-476

As the information was collected only through virtual interviews, triangulation could not be done, a limitation of the study.

Comment 2

Paragraph 1 (lines 45-52):

It would be helpful for the context of the research to have epidemiological measures both nationally and in the study site for the readers to get an idea of how big of a burden is malaria in Nepal and in the study site. The first paragraph of the introduction had numbers of malaria cases, but without any denominators it is difficult to put the information in context. I would suggest adding extra information such as annual malaria incidence. Having information on the proportion of P. vivax malaria will also make the result and discussion on treatment adherence more relevant.

Response 2

Thank-you for the suggestion. Additional epidemiological information has been added. Please refer to Page 2 and 3 Line 44-48.

Numbers of malaria cases in Nepal decreased from 11,000 in 2000 to 1,065 in 2019 and a low Annual Parasitic Incidence of 0.09 per 1,000 population at risk [3]. Among these 1065 malaria cases, 440 were indigenous (locally transmitted) and 625 were imported cases mostly from neighbouring India and countries in Africa [3]. Plasmodium vivax constituted 94.6% of the total cases in the country, whilst the remaining 5.4% were Plasmodium falciparum cases [3].

Comment 3

Methods

Research team (lines 105-110):

Because the researchers were not present in the study site during data collection, it is important to describe in more detail about the local research assistants, who they are, what were exactly their role in the study, and what were their relationship with the study participants.

Response 3

Thank-you for your suggestion. The details of the research assistants have been updated along with more information on their roles in the research. Please see page 5 line 113, 

…local research assistants with a public health background and currently working for the national malaria program at the study site. 

Page 6 line 131-135 

Based on these criteria, a mix of potential participants were identified by the principal researcher from different cohorts that included priests, health workers, past malaria patients, teachers and students to generate rich information. The RAs supported the recruitment of the participants at the study site and arranged the necessary logistics for conducting the virtual interviews. 

Page 7 line 151-152

Participants provided consent, both oral and written, before participating in the study and the written consent were collected by the RAs.

Comment 4

Participant eligibility and recruitment (lines 121-127):

I assume this research was done during the covid-19 pandemic so I expect some restrictions in contact between people. If this was the case at the study site, I think it is relevant to explain the method of approach and any non-participation.

Thank you for the comments. Additional information provided. Please refer to Page 6 line 131-135 

Based on these criteria, a mix of potential participants were identified by the principal researcher from different cohorts that included priests, health workers, past malaria patients, teachers and students to generate rich information. The RAs supported the recruitment of the participants at the study site and arranged the necessary logistics for conducting the virtual interviews. 

Comment 5

Data collection (lines 137-145):

In the discussion section (line 453-454) it was mentioned that the majority of interviews were done virtually. How many were done in person and how many were done virtually?

Where were the zoom interviews held in the study site? Is it at a clinic? At participants’ own houses? Or elsewhere?

Response 5

Thank-you for the comment. All the interviews were conducted by the principal investigator. Additional information added, please refer to page 7 line 149-151 …

While sixteen participants were interviewed in their own house, the remaining nine including health workers were interviewed in a meeting hall at the nearby health post.

... and correction made page 21 and 22 line 472-475

The inclusion of participants between 15 to 72 years of age allowed us to explore the malaria prevention and treatment practices and changes across generations. Due to the COVID -19 restrictions, the interviews had to be conducted virtually. 

Comment 6

In some of the quotes the participants mentioned their role as community health volunteers. It would be good if the results section started with a description of such characteristics. How many of the participants were community health volunteers? What about the other participants? What are their roles and positions in the society?

There is a paragraph in the discussion section about treatment adherence. Is there any data that can be added in the results section that might suggest malaria relapse or recurrence?

Response 6

Thank you for your comments and suggestions. Details of the participants have now been added. Please refer to page 8, line 174-176.

The results of the interviews (N=25) including illustrative quotes of the participants are presented. The participants included farmers (n= 13), priest/ traditional healers (n=2), health workers (n=2), school teacher (n= 2), students (n=2), self-employed/service (n=2) and local leaders (n=2).

Regarding relapses, only general information was sought and there is no published data on malaria relapse and recurrence in Nepal. However, five participants shared their experience of relapse which has been added in the text. Please refer to page 10 line 202.

However, five of the participants reported non-adherence to the treatment protocol. “They [villagers] take medicines for 4-5 days and feel better, and then they leave the medicine as it is. We [community health volunteers] and the health workers all went there [patient’s house], convinced them, and made them take the medicines” (female, FCHV).

Comment 7

Discussion

The lack of test of validity of the findings due to using only one type of data collection methods with no triangulation should be mentioned as one of the limitations.

Response 7

Thank you for the suggestion. This has now been added in the manuscript under limitation, please refer to page 22, line 475-476

As the information was collected through virtual interviews, triangulation could not be done, a limitation of the study.

Comment 8

Title and abstract

As not all readers would be familiar with places in Nepal, I personally think it might benefit the paper if the title and/or abstract mentions more descriptive words such as “remote” and/or “upper river valley.”

Response 8 

Thank-you for your suggestion. The title of the study has been changed accordingly please refer to Page1 Line 1

A Qualitative Study of Knowledge, Attitude and Perceptions towards Malaria Prevention among People Living in Rural Upper River Valleys of Nepal

Reviewer 2

Comment 1

Methods

Line 121 : Participant eligibility and recruitment : for in-depth interviews, participants are generally people who interact with many people in the community on a daily basis e.g. -head of districts, religious leaders, youth or women leaders etc.- as they are well informed on the population KAP… Pleased clarify if those people were included as participants. Also provide more details on the criteria used for the selection of participants.

Response 1

Thank you for your comments and suggestions. Details of the participants have now been added. Please refer to page 8, line 17r4-176.

The results of the interviews (N=25) including illustrative quotes of the participants are presented. The participants included farmers (n= 13), priest/ traditional healers (n=2), health workers (n=2), school teacher (n= 2), students (n=2), self-employed/service (n=2) and local leaders (n=2).

Comment 2

Line 130 “A semi-structured interview guide was developed based on similar research [17] and the researchers’ experience exploring the topic of malaria” Can the author add as additional file the questionnaire or the guide used for the interview.

Response 2

Thank you very much, we have included the domains of enquiry in English for the reviewer.

Comment 3

Was survey done during the rainy or dry season? Please clarify as this might affect perception or risk/attitudes toward malaria prevention and treatment.

Response 3

Thank you for your suggestion. Some information has been added, please refer to Page 7 line 146-147. The suggestion is an interesting concept and could be a potential area for further research.

One-on-one interviews were conducted between November and December 2020 (post monsoon) by the lead author (KA) with local support from Research Assistants

Comment 4

Results

The author need to be more precise on the number who responded over the number interviewed.

For instance Line 183 “Most of the participants knew that they could get anti-malarial medicine from…” How many please give the number

Another Line 190 “Only half of the participants mentioned that the government had conducted…” They interviewed 25 persons what do they consider as half is it 12 or 13 people please clarify. There are many places in the document where similar terms are used they need to check the whole result section and provide full details on the number of respondent.

See again below

Line 264 : “The majority of the participants (n=10) informed that washed nets were dried in a shaded area as recommended by the manufacturer” is 10 out of 25 interviewed represents the majority ?

also see line 271 “However, most of the participants reported experiencing several negative consequences of….” Wow many please give the precise number

Line 273 “Nearly half of the participants (n=11) shared their experiences of increased bed bugs after…”. Please check and correct accordingly.

Response 3

Thank you for your suggestion. The information has been added page 9 to 14 line 186-317 highlighted in blue. 

Comment 4

Line 296 : I don’t see the need of adding this figure 

Response 4

Thank you for the suggestion the figure has been removed.

Editor’s comments

Comment 1

Response

Thank-you for the suggestion, the manuscript has been revised using the PLOS One style template

Comment 2

Please include additional information regarding the survey or interview guide used in the study and ensure that you have provided sufficient details that others could replicate the analyses. For instance, if you developed a survey guide as part of this study and it is not under a copyright more restrictive than CC-BY, please include a copy, in both the original language and English, as Supporting Information.

Response 2

Thank you very much, we have included the domains of enquiry in English.

Comment 3

Thank you for stating the following financial disclosure: 

Response 3 

Thanks you for the information. This has been indicated in the cover letter attached with the revised submission. 

Comment 4

We note that Figure 2 in your submission contain copyrighted images. All PLOS content is published under the Creative Commons Attribution License (CC BY 4.0), which means that the manuscript, images, and Supporting Information files will be freely available online, and any third party is permitted to access, download, copy, distribute, and use these materials in any way, even commercially, with proper attribution. For more information, see our copyright guidelines: http://journals.plos.org/plosone/s/licenses-and-copyright.

Response 4

The photo was taken by the principal investigator during his visit to the study site on 25th April 2018 and holds the copyright himself. However, now figure 2 has been removed.

Comment 5

Additional Editor Comments:

Kindly add to the limitations section within the discussion as suggested by reviewer 1, and address all comments of both reviewers.

Response 5

Thank you for the suggestion. The limitation section has been revised with additional information based on reviewer’s suggestion. Please refer to Page 21 lines 475-476.

As the information was collected through virtual interviews, triangulation could not be done, a limitation of the study.

---

## [Decision Letter · Decision Letter 1]

3 Feb 2022

PONE-D-21-28469R1A qualitative study of knowledge, attitude and perceptions towards malaria prevention among people living in rural upper river valleys of NepalPLOS ONE

Dear Dr. Awasthi,

Thank you for submitting your manuscript to PLOS ONE. After careful consideration, we feel that it has merit but does not fully meet PLOS ONE’s publication criteria as it currently stands. Therefore, we invite you to submit a revised version of the manuscript that addresses the points raised during the review process.

Kindly have a native speaker familiar with the topic revise the entire manuscript for language.

We look forward to receiving your revised manuscript.

Kind regards,

Benedikt Ley, PhD

Academic Editor

PLOS ONE

Journal Requirements:

Reviewers' comments:

Reviewer's Responses to Questions

**Comments to the Author**

1. If the authors have adequately addressed your comments raised in a previous round of review and you feel that this manuscript is now acceptable for publication, you may indicate that here to bypass the “Comments to the Author” section, enter your conflict of interest statement in the “Confidential to Editor” section, and submit your "Accept" recommendation.

Reviewer #1: All comments have been addressed

Reviewer #3: (No Response)

2. Is the manuscript technically sound, and do the data support the conclusions?

Reviewer #1: Partly

Reviewer #3: Partly

3. Has the statistical analysis been performed appropriately and rigorously? 

Reviewer #1: N/A

Reviewer #3: N/A

4. Have the authors made all data underlying the findings in their manuscript fully available?

Reviewer #1: No

Reviewer #3: Yes

5. Is the manuscript presented in an intelligible fashion and written in standard English?

Reviewer #1: Yes

Reviewer #3: No

6. Review Comments to the Author

Reviewer #1: Dear author

Thank you for addressing the comments made in the previous review. However, there are some additional things for the discussion section that I believe would improve the quality of the manuscript.

1. There is a paragraph about treatment adherence (lines 335-352) in the discussion section. However, I do not see this being a prominent finding in the results section. I would suggest to put more data about treatment adherence issues in the result section to justify this paragraph or remove/reduce discussion about treatment adherence issue (perhaps as part of malaria treatment knowledge?).

2. Socioeconomic status (SES) was mentioned and discussed in the paragraph regarding malaria knowledge (lines 322-334). In my view, this is also relevant in the later paragraph discussing access to treatment (380-401). While this manuscript is not focused on intersectionality between different factors, I think it will benefit the manuscript to bring readers' attention to the complexity of malaria-related problem in this setting.

3. Virtual interview is mentioned as a limitation of this study. In the current world we are living in, virtual interview and even virtual ethnography might be explored more. Is there anything readers could learn from your experience doing the virtual interviews? In your opinion, is this method of data collection have any influence on the results? Would the results of the interview be any different (apart from lack of 'rich' data and ability to triangulate) had it been done in person? How has the study participants react to their involvement in a study that uses this kind of technology? What was the measures taken to ensure confidentiality and anonymity, since I assume the recorded version of the interviews include video showing the faces of the study participants? The answers to these questions might worth a paragraph in the discussion section.

4. In relation to point #3, I think a paragraph needs to be added in the methods section explaining confidentiality and anonymity, including who have access to the recorded file and where it is stored.

5. In the abstract, I would prefer to use "in-depth interviews" rather than "one-on-one interviews" (line 26) as that is the common term and is what is stated in the methods section (line 138). Or perhaps "virtual in-depth interviews"?

Reviewer #3: Review of Chan et al. 2021 Manuscript

General comments:

• Make sure your qualitative approach, methods, and analysis are all in line with one another.

• Consider using the Oxford comma throughout the manuscript.

• Would be good to have scientific editors review this for language.

• Explain the limitations in sharing raw data from manuscripts in the methods section.

Specific comments:

Abstract

Line 29: remove “from” before females and males.

Remove “females and males” on line 30 and put “female and male” before participants.

Line 31: “the complications” explain/specify complications from or of what.

Line 32: “were provided freely,” freely does not mean free, would change the word to “for free” or “free of charge”

Introduction

Line 45: remove “to” after contributes; add “The” before numbers and remove the “s” from number. � “The number of malaria cases in Nepal decreased…”

Line 47: remove “of” before “malaria prevention interventions”; add a comma before “including”; add “the” before “free distribution”

Line 51: “all supported by greater international funding.” � what is meant by “greater” with regards to international funding?

Line 52: change “the National Malaria Strategic Plan: to “its National Malaria Strategic Plan”

Line 54: remove “to” after “contributed”; after “Geographically” add “and topographically”

Line 55: remove “further” before “divided”

Line 57: before “experience” add “, which”

Line 58: add “cross border” before “movement”; add “from Nepal” before “to India”

Line 59: after “The valleys” should “along” be substituted with “between”? Is it in between the two geographies you describe?

Line 60: change “outbreak episodes” to “outbreaks” ; Would also describe some of the epidemiological characteristics of the outbreak, including species distribution,…...

Line 63: “a village of Khatyad..” changed to “a village in…”

Line 64: change “situated” to “located”; change “URV of” to “URV in”

Line 73: remove “the” after “associated with”;

Line 74: remove of communities -> Line 73+74: “Malaria is often considered a disease of the poor and has been associated with communities’ economic status and living conditions.”

Line 76: move “early” from before “health care” to after “health care”

Line 79: add a comma “,” after priorities’ change “expenditure” to “purchasing”; Line 79+80� “…, and purchasing bed nets and other malaria prevention tools is not a priority.”

Line 82: change “lead to the migration of villagers to endemic urban plains in Nepal…” to “lead to villagers migrating to malaria endemic urban plains in Nepal..” + would provide more description of migration—short-term? Long-term? Daily? Weekly? Monthly?

Line 81-83: explain why this movement exposes people to an increase of malaria transmission.

Line 84: change “The” at the beginning of the sentence to “Nepal’s”

Line 86: is the “its importance” referring to regular use of the LLINs or simply the importance of LLINs? If it’s the importance of the LLINs then “its” should be changed to “their.”

Line 89: add “season” after “monsoon”

Line 92: change “repeated episodes” to “several rounds”

Line 94: Replace “leading to refusal to allow” with “leading to households refusing to allow…” ; Line 94+95 Replace “challenges to these key pillars of malaria control” to “barriers to the success of let malaria control interventions,”

Line 95: add “the” between “explore” and “knowledge”; add “of” after “knowledge”; add “about” after “attitudes”

Line 96: remove comma “,” between “treatment” and “among”

Line 97: define “wards”

Materials and Methods Line 100: remove parenthesis “)” after “[21]”

Line 116: add “season” after “monsoon” ; change “during winter” to “in the winter”

Line 120: add a comma “,” after “’medical’”

Line 123: remove “being” before “between the ages…”

Line 125: add “their” before “roles” ; add “health related” between “in” and “decision-making” + remove “for health care”

Would be good to mention the roles/occupation of those interviewed. Also, in the results community health workers are quoted, should mention that in the paragraph on participant eligibility and recruitment. Especially, given that a phenomenological approach was taken to the qualitative research, would have to explain why health care community workers were included in the interviews if this was about high-risk population/people living in the area and their experience with malaria, its treatment, and prevention.

Line 139: remove “the” before “COVID 19 travel restrictions”

Line 140: change “zoom platform” to “Zoom video conferencing”

Line 150: add “for” after “searching”

Line 155: remove “,” comma after “organized” + add “and” after “organized”

Line 157: change “elaborate” to “develop”

Line 153: given that you are conducting a phenomenological qualitative study and you used open coding, I would add a justification for the open coding here, perhaps that you are conducting exploratory research into ideas and concepts that have not been explored in this area before.

Results

Figure 1: I would change the order to have the “3. Availability and Practices” third on the right, so that it reads 1,2,3 from left to right.

Line 170: change “unclear of” to “unclear about”

Line 171: “Among them” who is the “them” referring to? If it is the participants, it is not needed.

Line 178: add “malaria” before “knowledge”

Line 181: “weakness”: what is meant by weakness here? Might be good to add an illustration if their conceptualization of weakness was explored.

Line 184: describes participants having to two different types of tablets for 14 to 15 days. Might need to comment here that this is the common treatment for vivax malaria or a least for the specific type of malaria the participants had.

Line 189/190: does the female FCHV say why her patient threw away the medicine, if so, might be good to include here.

Line 204: remove semi-collin “;”

Line 211-219: description of what happened in the “past several years” -> is this description something that is still happening or is it a thing of the past? If it is no longer happening, how and why did it change?

Line 220: change first two sentences to: “Traditional management of malaria was described by participants as visits to traditional healers known as ‘Dhamis’ or ‘Jhakris’. However, the use of traditional healers did not seem common. The participants reported…”

Line 226: look like a main header, should be under the ‘choices and preferences’ header right?

Line 239: remove “and journey of getting” and replace with “of the journey to get”

Line 242: move the USD cost before the ending quotation mark.

Line 244: change “the treatment options” to “where treatment is sought”

Line 245: change “for” to “regarding” + add “matters” after “health”

Line 252: division of what? | remove “the” before “LLINS”

Line 255: remove “round” after “year”

Line 258: add a comma “,” after detergents and add “baking” after soda otherwise need to explain what is meant by soda here.

Line 261: add a comma “,” between “nets” and “and”

Line 265: change “informed” to “mentioned”

Line 272: change “in the households” to “in their houses.”

Line 283: add “also” after “participants” ; change “the family business of bee-farming” to “bee-farming family businesses”

Line 288: remove comma “,” after bees

Line 293: “several participants in the community” -> are these interview participants?

Discussion Line 300: change “knowledge on” to “knowledge of” ; “However,” might not be necessary to start the second sentence on that line.

Line 303: “in the area” � what is meant by in the area? Which area?

Line 304: add “malaria” before “prevention and treatment”

Line 312: what is consistent with research by Yadaw et al? “90% of respondents” -> whose respondents are these—which study is this from. Not clear.

Line 314: change “the higher SES” to “those of higher SES”

Line 316: how are you defining lower SES group? What want to consider saying “are of low socio-economic status, but make sure to define.

Line 320: add “treatment” before “protocol”

Line 329: is this how the patients described their side effects? Anorexia and allergic reactions. Seems like pretty technical language.

Line 330: the trial talked about recurrence, not relapse. Would also add in the dosage information from the trial.

Line 350/351: “Malaria treatment is free in public health facilities across Nepal, a reason that could explain that preference.” Isn’t it the case that participants provided the reasoning for their preference of public health facilities? Would be careful how you formulate this sentence.

Line 352/353: description of traditional healing as still being an option. Is not quite in alignment with how you report it in the result section.

Line 354: change “acceptance to” to “acceptance of”

Line 355: change “(western)” “(western-style)”

Line 363: change “limitations in” to “limitations of”

Line 368: remove “completeness”

Line 369: change to “, due to weak monitoring and lack of commercially available Primaquine in Nepal [31].”

Line 373: countries like Bangladesh or actually Bangladesh? + remove “in” after “among”

Line 374: change “have no reporting access to the national…” to “cannot report to the national….”

Line 375: change “resulting in missed cases” to “resulting in the surveillance system missing cases.”

Line 377-380: feels like this sentence comes out nowhere. Would provide the findings this recommendation is based on first then provide the recommendation.

Line 382: add “at” between “treatment” and “facilities”

Line 383: change “this” to “Such accessibility and availability”

Line 384: explain why it would be crucial to prevent local transmission or outbreaks.

Line 389: Add comma “,” after “insurance system” and after “(OPP)”

Line 390: add “seeking” after “health care”

Line 892: remove “the” before “physical” + remove “s” at the end of “conditions” + change “and” to “or” before “old age”

Line 394: change “transport facilities” to “transport infrastructure”

Line 398: change “saving them” to “mitigating”

Line 400: change “for a” after “known” to “as a”

Line 401: add a comma “,” after “society” + change “taken” to “made” after “were”

Lines 407-409: where is the description?

Line 418: Remove “The” before “LLINs” change “a factor for” to “a factor in”

Line 420: add “a” before “high risk of malaria” + remove comma “,” after “malaria”

Line 421: remove “the” before “private”

Line 422: change “access to” to “access for”

Line 423: add “Additionally,” before “Normal nets”

Line 426: add “baking” before “soda” + change “for washing the LLINs” to “to wash LIINs” + remove the comma “,” after “LLINs”.

Line 428: change “for washing” to “to wash”

Line 431: change “negative consequences of IRS” to “unintended negative consequences from IRS”

Line 433: change “barriers for” to “barriers to”

Line 434: change “shifting” to “moving” before “the beehives” + add an “s” to “outdoor” + change “in” to “to” after “outdoor”

Line 439: add “from our findings.” after “differed” + end the sentence there and start a new one with “Barriers identified by respondents in xxxx included people believing it increased rodents….” -> need to also add context to these barriers i.e., location.

Line 443: examples of political factors.

Line 443: first word, change “for building” to “creating”

Line 445: add “described by our participants” after “negative experiences”

Line 446: change “albeit” to “while”

Limitations

Line 450: “the local community” -> specify which local community.

Line 452: remove “the” before “malaria prevention”

Line 453: remove “the” before “COVID-19 restrictions”

Line 455: add “malaria” before “endemic”

Line 456: change “people on” to “people relating to”

Conclusion

Line 463: “people in rural areas to seek alternative care from traditional healers and private medicals.” -> This isn’t in agreement with your results. Would make this more nuanced so it is in line with the reporting of results.

7. PLOS authors have the option to publish the peer review history of their article (what does this mean?). If published, this will include your full peer review and any attached files.

Reviewer #1: No

Reviewer #3: No

---

## [Author Response · Author response to Decision Letter 1]

2 Mar 2022

Reviewers Response

Manuscript title: Title: A Qualitative Study of Knowledge, Attitudes and Perceptions towards Malaria Prevention among People Living in Khatyad Rural Municipality of Mugu, Nepal

Dear reviewers, thank you for your comments and suggestions which have been addressed and track changed in the manuscript for your kind perusal. We would also like to acknowledge the effort and time that you have provided to go through our manuscript. 

Reviewer 1

Comment 1

There is a paragraph about treatment adherence (lines 335-352) in the discussion section. However, I do not see this being a prominent finding in the results section. I would suggest to put more data about treatment adherence issues in the result section to justify this paragraph or remove/reduce discussion about treatment adherence issue (perhaps as part of malaria treatment knowledge?).

Response 1

Thank you for your comments and suggestion. There is a paragraph on treatment adherence in the results section please refer to page 11 line 216-222. Some additional information has also been added. Knowledge and practice of drug adherence was an important finding in the study, particularly due to the fact that Plasmodium vivax is the predominant species in the area. 

Comment 2

Socioeconomic status (SES) was mentioned and discussed in the paragraph regarding malaria knowledge (lines 322-334). In my view, this is also relevant in the later paragraph discussing access to treatment (380-401). While this manuscript is not focused on intersectionality between different factors, I think it will benefit the manuscript to bring readers' attention to the complexity of malaria-related problem in this setting.

Response 2

Thank you for your comments. As you have correctly indicated, the paper is not focused on intersectionality. The study explored the underlying factors of how SES may be associated with poor knowledge and reluctance to seek appropriate health care (please refer to page 16 line 345- 346). 

Comment 3

Virtual interview is mentioned as a limitation of this study. In the current world we are living in, virtual interview and even virtual ethnography might be explored more. Is there anything readers could learn from your experience doing the virtual interviews? In your opinion, is this method of data collection have any influence on the results? Would the results of the interview be any different (apart from lack of 'rich' data and ability to triangulate) had it been done in person? How has the study participants react to their involvement in a study that uses this kind of technology? What was the measures taken to ensure confidentiality and anonymity, since I assume the recorded version of the interviews include video showing the faces of the study participants? The answers to these questions might worth a paragraph in the discussion section.

Response 3

Thank you very much for this is an interesting observation. The virtual interviews compromised the discussion slightly due to poor network, however we still felt the interaction was successful and very similar to conducting a face to face interview. Please refer to page 23 line 508-511. Only audio recordings were saved (page 8 line 170) and anonymity was maintained by using pseudonyms and removing any personal identifiers (page 8 line 171-173).

Comment 4 

In relation to point #3, I think a paragraph needs to be added in the methods section explaining confidentiality and anonymity, including who have access to the recorded file and where it is stored.

Response 4

Thank you very much for your comments. This research has been approved by two ethical research committees, (i.e. NHRC ethics of Nepal and HREC of Curtin University) and anonymity and confidentiality have been strictly adhered to. All the raw data has been securely stored in the Curtin university HREC repository and this has been added in the manuscript. Please refer to page 8 line 172. 

Comment 5 

In the abstract, I would prefer to use "in-depth interviews" rather than "one-on-one interviews" (line 26) as that is the common term and is what is stated in the methods section (line 138). Or perhaps "virtual in-depth interviews"?

Response 5

Thank you for the suggestion. The one-on-one interviews have been changed to in-depth interviews as suggested (page 2 line 29 and page 8 line 162)

Reviewer 3

Section Reviewers Comments Response

Thank you for your meticulous effort in providing editorial feedback on the manuscript which is highly appreciated. We have corrected the grammatical errors and punctuations as per the suggestion.

Abstract Line 29: remove “from” before females and males. 

Remove “females and males” on line 30 and put “female and male” before participants. Thank you for your suggestion. Correction has been made as suggested. Please refer to page 2 line 30

 Line 31: “the complications” explain/specify complications from or of what. Thank you for your suggestion. Correction has been made as suggested. Please refer to page 2 line 32

 Line 32: “were provided freely,” freely does not mean free, would change the word to “for free” or “free of charge” Thank you for your suggestion. Correction has been made as suggested. Please refer to page 2 line 33

Introduction Line 45: remove “to” after contributes; add “The” before numbers and remove the “s” from number. � “The number of malaria cases in Nepal decreased…” Thank you for your suggestion. Correction has been made as suggested. Please refer to page 3 line 46

 Line 47: remove “of” before “malaria prevention interventions”; add a comma before “including”; add “the” before “free distribution” Thank you for your suggestion. Correction has been made as suggested. Please refer to page 3 line 53 

 Line 51: “all supported by greater international funding.” � what is meant by “greater” with regards to international funding? Thank you for your suggestion. Correction has been made as suggested. Please refer to page 3 line 56

 Line 52: change “the National Malaria Strategic Plan: to “its National Malaria Strategic Plan” Thank you for your suggestion. Correction has been made as suggested. Please refer to page 3 line 58

 Line 54: remove “to” after “contributed”; after “Geographically” add “and topographically” Thank you for your suggestion. Correction has been made as suggested. Please refer to page 3 line 61,62

 Line 55: remove “further” before “divided” Thank you for your suggestion. Correction has been made as suggested. Please refer to page 3 line 62

 Line 57: before “experience” add “, which” Thank you for your suggestion. Correction has been made as suggested. Please refer to page 3 line 64

 Line 58: add “cross border” before “movement”; add “from Nepal” before “to India” Thank you for your suggestion. Correction has been made as suggested. Please refer to page 3 line 65

 Line 59: after “The valleys” should “along” be substituted with “between”? Is it in between the two geographies you describe? Thank you for your suggestion. Correction has been made as suggested. Please refer to page 3 line 66

 Line 60: change “outbreak episodes” to “outbreaks” ; Would also describe some of the epidemiological characteristics of the outbreak, including species distribution,…... Thank you for your suggestion. Correction has been made as suggested. Additional information added to bring clarity to cases. Please refer to page 3 line 68 and page 4 line 70

 Line 63: “a village of Khatyad.” changed to “a village in…” Thank you for your suggestion. Correction has been made as suggested. Please refer to page 4 line 71

 Line 64: change “situated” to “located”; change “URV of” to “URV in” Thank you for your suggestion. Correction has been made as suggested. Please refer to page 4 line 72

 Line 73: remove “the” after “associated with”; Thank you for your suggestion. Correction has been made as suggested. Please refer to page 4 line 81

 Line 74: remove of communities � Line 73+74: “Malaria is often considered a disease of the poor and has been associated with communities’ economic status and living conditions.” Thank you for your suggestion. Correction has been made as suggested. Please refer to page 4 line 82

 Line 76: move “early” from before “health care” to after “health care” Thank you for your suggestion. Correction has been made as suggested. Please refer to page 4 line 84-85

 Line 79: add a comma “,” after priorities’ change “expenditure” to “purchasing”; Line 79+80� “…, and purchasing bed nets and other malaria prevention tools is not a priority.” Thank you for your suggestion. Correction has been made as suggested. Please refer to page 4 line 82, 87

 Line 82: change “lead to the migration of villagers to endemic urban plains in Nepal…” to “lead to villagers migrating to malaria endemic urban plains in Nepal..” + would provide more description of migration—short-term? Long-term? Daily? Weekly? Monthly? Thank you for the suggestion. The editing has been done as per advice and addition clarity in information has been added for the case characteristics. Please refer to page 4 and 5 line 89, 91-93

 Line 81-83: explain why this movement exposes people to an increase of malaria transmission. Thank you for your suggestion. Correction has been made as suggested. Please refer to page 4 and 5 line 91-93

 Line 84: change “The” at the beginning of the sentence to “Nepal’s” Thank you for your suggestion. Correction has been made as suggested. Please refer to page 5 line 95

 Line 86: is the “its importance” referring to regular use of the LLINs or simply the importance of LLINs? If it’s the importance of the LLINs then “its” should be changed to “their.” Thank you for your suggestion. Correction has been made as suggested. Please refer to page 5 line 97

 Line 89: add “season” after “monsoon” Thank you for your suggestion. Correction has been made as suggested. Please refer to page 5 line 100

 Line 92: change “repeated episodes” to “several rounds” Thank you for your suggestion. Correction has been made as suggested. Please refer to page 5 line 104

 Line 94: Replace “leading to refusal to allow” with “leading to households refusing to allow…” ; Line 94+95 Replace “challenges to these key pillars of malaria control” to “barriers to the success of let malaria control interventions,” Thank you for your suggestion. Correction has been made as suggested. Please refer to page 5 line 105-107

 Line 95: add “the” between “explore” and “knowledge”; add “of” after “knowledge”; add “about” after “attitudes” Thank you for your suggestion. Correction has been made as suggested. Please refer to page 5 line 108

 Line 96: remove comma “,” between “treatment” and “among” Thank you for your suggestion. Correction has been made as suggested. Please refer to page 5 line 109

 Line 97: define “wards” Thank you for your suggestion. Correction has been made as suggested. Please refer to page 5 line 110

Materials and Methods Line 100: remove parenthesis “)” after “[21]” Thank you for your suggestion. Correction has been made as suggested. Please refer to page 6 line 113

 Line 116: add “season” after “monsoon” ; change “during winter” to “in the winter” Thank you for your suggestion. Correction has been made as suggested. Please refer to page 6 line 132

 Line 120: add a comma “,” after “’medical’” Thank you for your suggestion. Correction has been made as suggested. Please refer to page 6 line 135

 Line 123: remove “being” before “between the ages…” Thank you for your suggestion. Correction has been made as suggested. Please refer to page 7 line 139

 Line 125: add “their” before “roles” ; add “health related” between “in” and “decision-making” + remove “for health care” Thank you for your suggestion. Correction has been made as suggested. Please refer to page 7 line 141

 Would be good to mention the roles/occupation of those interviewed. Also, in the results community health workers are quoted, should mention that in the paragraph on participant eligibility and recruitment. Especially, given that a phenomenological approach was taken to the qualitative research, would have to explain why health care community workers were included in the interviews if this was about high-risk population/people living in the area and their experience with malaria, its treatment, and prevention. Thank you for your suggestion. Additional information has been added as suggested. Please refer to page 7 line 142-146

 Line 139: remove “the” before “COVID 19 travel restrictions” Thank you for your suggestion. Correction has been made as suggested. Please refer to page 8 line 164

 Line 140: change “zoom platform” to “Zoom video conferencing” Thank you for your suggestion. Correction has been made as suggested. Please refer to page 8 line 165

 Line 150: add “for” after “searching” Thank you for your suggestion. Correction has been made as suggested. Please refer to page 8 line 178

 Line 155: remove “,” comma after “organized” + add “and” after “organized” Thank you for your suggestion. Correction has been made as suggested. Please refer to page 8 line 184

 Line 157: change “elaborate” to “develop” Thank you for your suggestion. Correction has been made as suggested. Please refer to page 9 line 186

 Line 153: given that you are conducting a phenomenological qualitative study and you used open coding, I would add a justification for the open coding here, perhaps that you are conducting exploratory research into ideas and concepts that have not been explored in this area before. Thank you for your suggestion. Additional information has been added as per the suggestion. Please refer to page 8 line 181

Results Figure 1: I would change the order to have the “3. Availability and Practices” third on the right, so that it reads 1,2,3 from left to right. Thank you for your suggestion. Correction has been made to the figures suggested. Please refer to page 10 line 198-199

 Line 170: change “unclear of” to “unclear about” Thank you for your suggestion. Correction has been made as suggested. Please refer to page 10 line 203

 Line 171: “Among them” who is the “them” referring to? If it is the participants, it is not needed. Thank you for your suggestion. Correction has been made as suggested. Please refer to page 10 line 204

 Line 178: add “malaria” before “knowledge” Thank you for your suggestion. Correction has been made as suggested. Please refer to page 11 line 211

 Line 181: “weakness”: what is meant by weakness here? Might be good to add an illustration if their conceptualization of weakness was explored. Thank you for your suggestion. Correction has been made as suggested. Please refer to page 11 line 214

 Line 184: describes participants having to two different types of tablets for 14 to 15 days. Might need to comment here that this is the common treatment for vivax malaria or a least for the specific type of malaria the participants had. Thank you for your suggestion. Correction has been made as suggested. Please refer to page 11 line 218

 Line 189/190: does the female FCHV say why her patient threw away the medicine, if so, might be good to include here. Thank you for your suggestion. Some additional information has been added as per the suggestion. Please refer to page 11 line 221,222

 Line 204: remove semi-colin “;” Thank you for your suggestion. Correction has been made as suggested. Please refer to page 12 line 240

 Line 211-219: description of what happened in the “past several years” � is this description something that is still happening or is it a thing of the past? If it is no longer happening, how and why did it change? Thank you for your suggestion. Correction has been made as suggested. Please refer to page 12 line 247 

 Line 220: change first two sentences to: “Traditional management of malaria was described by participants as visits to traditional healers known as ‘Dhamis’ or ‘Jhakris’. However, the use of traditional healers did not seem common. The participants reported…” Thank you for your suggestion. Correction has been made as suggested. Please refer to page 13 line 256-258

 Line 226: look like a main header, should be under the ‘choices and preferences’ header right? Thank you for your suggestion. Correction has been made as suggested. Please refer to page 13 line 264

 Line 239: remove “and journey of getting” and replace with “of the journey to get” Thank you for your suggestion. Correction has been made as suggested. Please refer to page 14 line 277

 Line 242: move the USD cost before the ending quotation mark. Thank you for your suggestion. Correction has been made as suggested. Please refer to page 14 line 280

 Line 244: change “the treatment options” to “where treatment is sought” Thank you for your suggestion. Correction has been made as suggested. Please refer to page 14 line 282

 Line 245: change “for” to “regarding” + add “matters” after “health” Thank you for your suggestion. Correction has been made as suggested. Please refer to page 14 line 284

 Line 252: division of what? | remove “the” before “LLINS” Thank you for your suggestion. Correction has been made as suggested. Please refer to page 14 line 291. EDCD is a proper noun and is a section under the Ministry of Health in Nepal that oversees communicable and non-communicable diseases.

 Line 255: remove “round” after “year” Thank you for your suggestion. Correction has been made as suggested. Please refer to page 14 line 294

 Line 258: add a comma “,” after detergents and add “baking” after soda otherwise need to explain what is meant by soda here. Thank you for your suggestion. Correction has been made as suggested. Please refer to page 14 line 298

 Line 261: add a comma “,” between “nets” and “and” Thank you for your suggestion. Correction has been made as suggested. Please refer to page 15 line 302

 Line 265: change “informed” to “mentioned” Thank you for your suggestion. Correction has been made as suggested. Please refer to page 15 line 305

 Line 272: change “in the households” to “in their houses.” Thank you for your suggestion. Correction has been made as suggested. Please refer to page 15 line 312

 Line 283: add “also” after “participants” ; change “the family business of bee-farming” to “bee-farming family businesses” Thank you for your suggestion. Correction has been made as suggested. Please refer to page 16 line 323

 Line 288: remove comma “,” after bees Thank you for your suggestion. Correction has been made as suggested. Please refer to page 16 line 328

 Line 293: “several participants in the community” � are these interview participants? Thank you for your suggestion. These are the interview participants. Correction has been made as suggested. Please refer to page 16 line 333

Discussion Line 300: change “knowledge on” to “knowledge of” ; “However,” might not be necessary to start the second sentence on that line. Thank you for your suggestion. Correction has been made as suggested. Please refer to page 16 line 339

 Line 303: “in the area” � what is meant by in the area? Which area? Thank you for your suggestion. Correction has been made as suggested. Please refer to page 16 line 342

 Line 304: add “malaria” before “prevention and treatment” Thank you for your suggestion. Correction has been made as suggested. Please refer to page 16 line 343

 Line 312: what is consistent with research by Yadav et al? “90% of respondents” � whose respondents are these—which study is this from. Not clear. Thank you for your suggestion. Correction has been made as suggested. Please refer to page 17 line 353

 Line 314: change “the higher SES” to “those of higher SES” Thank you for your suggestion. Correction has been made as suggested. Please refer to page 17 line 355

 Line 316: how are you defining lower SES group? What want to consider saying “are of low socio-economic status, but make sure to define. Thank you for your suggestion. The lower SES has been rephrased as per the suggestion. Please refer to page line 6 line 129 under methods. The referred study Yadav et al. conducted in Rajasthan of India does not clearly state the SES categories.

 Line 320: add “treatment” before “protocol” Thank you for your suggestion. Correction has been made as suggested. Please refer to page 17 line 363

 Line 329: is this how the patients described their side effects? Anorexia and allergic reactions. Seems like pretty technical language. Thank you for your suggestion. Correction has been made as suggested. Please refer to page 18 line 371-372

 Line 330: the trial talked about recurrence, not relapse. Would also add in the dosage information from the trial. Thank you for your suggestion. Correction has been made as suggested and additional information on dosage has been added. Please refer to page 18 line 373-375

 Line 350/351: “Malaria treatment is free in public health facilities across Nepal, a reason that could explain that preference.” Isn’t it the case that participants provided the reasoning for their preference of public health facilities? Would be careful how you formulate this sentence. Thank you for your suggestion. Correction has been made as suggested. Please refer to page 19 line 395

 Line 352/353: description of traditional healing as still being an option. Is not quite in alignment with how you report it in the result section. Thank you for your suggestion. Correction has been made as suggested. Please refer to page 19 line 396-398

 Line 354: change “acceptance to” to “acceptance of” Thank you for your suggestion. Correction has been made as suggested. Please refer to page 19 line 399

 Line 355: change “(western)” “(western-style)” Thank you for your suggestion. Correction has been made as suggested. Please refer to page 19 line 399

 Line 363: change “limitations in” to “limitations of” Thank you for your suggestion. Correction has been made as suggested. Please refer to page 19 line 408 

 Line 368: remove “completeness” Thank you for your suggestion. Correction has been made as suggested. Please refer to page 19 line 413

 Line 369: change to “, due to weak monitoring and lack of commercially available Primaquine in Nepal [31].” Thank you for your suggestion. Correction has been made as suggested. Please refer to page 19 line 414

 Line 373: countries like Bangladesh or actually Bangladesh? + remove “in” after “among” Thank you for your suggestion. Correction has been made as suggested. Please refer to page 20 line 418

 Line 374: change “have no reporting access to the national…” to “cannot report to the national….” Thank you for your suggestion. Correction has been made as suggested. Please refer to page 20 line 419

 Line 375: change “resulting in missed cases” to “resulting in the surveillance system missing cases.” Thank you for your suggestion. Correction has been made as suggested. Please refer to page 20 line 420-421

 Line 377-380: feels like this sentence comes out nowhere. Would provide the findings this recommendation is based on first then provide the recommendation. Thank you for your suggestion. This has been removed. Please refer to page 20 line 429-433

 Line 382: add “at” between “treatment” and “facilities” Thank you for your suggestion. Correction has been made as suggested. Please refer to page 20 line 428

 Line 383: change “this” to “Such accessibility and availability” Thank you for your suggestion. Correction has been made as suggested. Please refer to page 20 line 429

 Line 384: explain why it would be crucial to prevent local transmission or outbreaks. Thank you for your suggestion. Correction has been made as suggested. Please refer to page 20 line 430-433

 Line 389: Add comma “,” after “insurance system” and after “(OPP)” Thank you for your suggestion. Correction has been made as suggested. Please refer to page 20 line 438

 Line 390: add “seeking” after “health care” Thank you for your suggestion. Correction has been made as suggested. Please refer to page 20 line 439

 Line 892: remove “the” before “physical” + remove “s” at the end of “conditions” + change “and” to “or” before “old age” Thank you for your suggestion. Correction has been made as suggested. Please refer to page 21 line 441

 Line 394: change “transport facilities” to “transport infrastructure” Thank you for your suggestion. Correction has been made as suggested. Please refer to page 21 line 443

 Line 398: change “saving them” to “mitigating” Thank you for your suggestion. Correction has been made as suggested. Please refer to page 21 line 447 

 Line 400: change “for a” after “known” to “as a” Thank you for your suggestion. Correction has been made as suggested. Please refer to page 21 line 434 

 Line 401: add a comma “,” after “society” + change “taken” to “made” after “were” Thank you for your suggestion. Correction has been made as suggested. Please refer to page 21 line 450-451

 Lines 407-409: where is the description? Thank you for your suggestion. The description is provided in the paragraph starting page 21 line 452 and 459 

 Line 418: Remove “The” before “LLINs” change “a factor for” to “a factor in” Thank you for your suggestion. Correction has been made as suggested. Please refer to page 22 line 469

 Line 420: add “a” before “high risk of malaria” + remove comma “,” after “malaria” Thank you for your suggestion. Correction has been made as suggested. Please refer to page 22 line 471

 Line 421: remove “the” before “private” Thank you for your suggestion. Correction has been made as suggested. Please refer to page 22 line 472

 Line 422: change “access to” to “access for” Thank you for your suggestion. Correction has been made as suggested. Please refer to page 22 line 473

 Line 423: add “Additionally,” before “Normal nets” Thank you for your suggestion. Correction has been made as suggested. Please refer to page 22 line 474

 Line 426: add “baking” before “soda” + change “for washing the LLINs” to “to wash LIINs” + remove the comma “,” after “LLINs”. Thank you for your suggestion. Correction has been made as suggested. Please refer to page 22 line 477

 Line 428: change “for washing” to “to wash” Thank you for your suggestion. Correction has been made as suggested. Please refer to page 22 line 479

 Line 431: change “negative consequences of IRS” to “unintended negative consequences from IRS” Thank you for your suggestion. Correction has been made as suggested. Please refer to page 22 line 483

 Line 433: change “barriers for” to “barriers to” Thank you for your suggestion. Correction has been made as suggested. Please refer to page 22 line 485

 Line 434: change “shifting” to “moving” before “the beehives” + add an “s” to “outdoor” + change “in” to “to” after “outdoor” Thank you for your suggestion. Correction has been made as suggested. Please refer to page 22 line 486, 487

 Line 439: add “from our findings.” after “differed” + end the sentence there and start a new one with “Barriers identified by respondents in xxxx included people believing it increased rodents….” � need to also add context to these barriers i.e., location. Thank you for your suggestion. Correction has been made as suggested. Please refer to page 23 line 492,493

 Line 443: examples of political factors. Thank you for your suggestion. Correction has been made as suggested. Please refer to page 23 line 497

 Line 443: first word, change “for building” to “creating” Thank you for your suggestion. Correction has been made as suggested. Please refer to page 23 line 498

 Line 445: add “described by our participants” after “negative experiences” Thank you for your suggestion. Correction has been made as suggested. Please refer to page 23 line 500

 Line 446: change “albeit” to “while” Thank you for your suggestion. Correction has been made as suggested. Please refer to page 23 line 501

Limitations Line 450: “the local community” � specify which local community. Thank you for your suggestion. Correction has been made as suggested. Please refer to page 23 line 504

 Line 452: remove “the” before “malaria prevention” Thank you for your suggestion. Correction has been made as suggested. Please refer to page 23 line 507

 Line 453: remove “the” before “COVID-19 restrictions” Thank you for your suggestion. Correction has been made as suggested. Please refer to page 23 line 508

 Line 455: add “malaria” before “endemic” Thank you for your suggestion. Correction has been made as suggested. Please refer to page 24 line 514

 Line 456: change “people on” to “people relating to” Thank you for your suggestion. Correction has been made as suggested. Please refer to page 24 line 515

Conclusion People in rural areas to seek alternative care from traditional healers and private medicals.” -> This isn’t in agreement with your results. Would make this more nuanced so it is in line with the reporting of results. Thank you for your suggestion. Correction has been made as suggested. Please refer to page 24 line 521

---

## [Editor Report · Decision Letter 2]

4 Mar 2022

A qualitative study of knowledge, attitudes and perceptions towards malaria prevention among people living in rural upper river valleys of Nepal

PONE-D-21-28469R2

Dear Dr. Awasthi,

We’re pleased to inform you that your manuscript has been judged scientifically suitable for publication and will be formally accepted for publication once it meets all outstanding technical requirements.

Kind regards,

Benedikt Ley, PhD

Academic Editor

PLOS ONE

---

## [Editor Report · Acceptance letter]

10 Mar 2022

PONE-D-21-28469R2 

A Qualitative Study of Knowledge, Attitudes and Perceptions towards Malaria Prevention among People Living in Rural Upper River Valleys of Nepal 

Dear Dr. Awasthi:

I'm pleased to inform you that your manuscript has been deemed suitable for publication in PLOS ONE. Congratulations! Your manuscript is now with our production department. 

Kind regards, 

on behalf of

Dr Benedikt Ley 

Academic Editor

PLOS ONE